# Whole Exome Sequencing of a Patient with a Milder Phenotype of Xeroderma Pigmentosum Group C

**DOI:** 10.3390/medicina59040699

**Published:** 2023-04-03

**Authors:** Ji-In Seo, Chikako Nishigori, Jung Jin Ahn, Jae Young Ryu, Junglok Lee, Mu-Hyoung Lee, Su Kang Kim, Ki-Heon Jeong

**Affiliations:** 1Department of Dermatology, College of Medicine, Kyung Hee University, Seoul 02447, Republic of Korea; 2Division of Dermatology, Internal Related, Graduate School of Medicine, Kobe University, Kobe 653-0002, Japan; 3Department of Oral Anatomy and Developmental Biology, Graduate School, Kyung Hee University, Seoul 02447, Republic of Korea; 4Department of Medicine, Graduate School, Kyung Hee University, Seoul 02447, Republic of Korea; 5Department of Biomedical Laboratory Science, Catholic Kwandong University, Gangneung 25601, Republic of Korea

**Keywords:** xeroderma pigmentosum group C, whole exome sequencing, bioinformatics analysis, XPC

## Abstract

A 17-year-old female Korean patient (XP115KO) was previously diagnosed with Xeroderma pigmentosum group C (XPC) by Direct Sanger sequencing, which revealed a homozygous nonsense mutation in the *XPC* gene (rs121965088: c.1735C > T, p.Arg579Ter). While rs121965088 is associated with a poor prognosis, our patient presented with a milder phenotype. Hence, we conducted whole-exome sequencing in the patient and her family members to detect coexisting mutations that may have resulted in a milder phenotype of rs121965088 through genetic interaction. *Materials and Methods*: the whole-exome sequencing analysis of samples obtained from the patient and her family members (father, mother, and brother) was performed. To identify the underlying genetic cause of XPC, the extracted DNA was analyzed using Agilent’s SureSelect XT Human All Exon v5. The functional effects of the resultant variants were predicted using the SNPinfo web server, and structural changes in the XPC protein using the 3D protein modeling program SWISS-MODEL. *Results*: Eight biallelic variants, homozygous in the patient and heterozygous in her parents, were detected. Four were found in the *XPC* gene: one nonsense variant (rs121965088: c.1735C > T, p.Arg579Ter) and three silent variants (rs2227998: c.2061G > A, p. Arg687Arg; rs2279017: c.2251-6A > C, intron; rs2607775: c.-27G > C, 5′UTR). The remaining four variants were found in non-XP genes, including one frameshift variant [rs72452004 of olfactory receptor family 2 subfamily T member 35 (*OR2T35*)], three missense variants [rs202089462 of ALF transcription elongation factor 3 (*AFF3*), rs138027161 of TCR gamma alternate reading frame protein (*TARP*), and rs3750575 of annexin A7 (*ANXA7*)]. *Conclusions*: potential candidates for genetic interactions with rs121965088 were found. The rs2279017 and rs2607775 of XPC involved mutations in the intron region, which affected RNA splicing and protein translation. The genetic variants of *AFF3*, *TARP*, and *ANXA7* are all frameshift or missense mutations, inevitably disturbing the translation and function of the resultant proteins. Further research on their functions in DNA repair pathways may reveal undiscovered cellular relationships within xeroderma pigmentosum.

## 1. Introduction

Xeroderma pigmentosum (XP) is an autosomal recessive genodermatosis that is associated with DNA repair defects. The disease comprises eight complementation groups (XP-A to XP g and XP-V), which represent functional deficiencies in different genes [damage specific DNA binding protein 2 (*DDB2*), ERCC excision repair, TFIIH core complex helicase subunit (*ERCC1 to 5*), DNA polymerase eta (*POLH*), XPA, DNA damage recognition and repair factor (*XPA)*, and XPC complex subunit, DNA damage recognition and repair factor (*XPC*)]. Proteins encoded by *DDB2*, *ERCC1* to *5*, *XPA*, and *XPC* participate in the nucleotide excisional repair (NER) system, whereas those by *POLH* are in translation with DNA synthesis. Depending on the complementation groups, XP patients demonstrate a great variety of clinical features and prognoses [1,2,3,4]. From mild manifestations such as lentiginous pigmentation, photophobia, and conjunctivitis, the disease may progress into lethal conditions such as cutaneous neoplasm, ocular surface cancer, internal organ malignancy, and epileptic seizure. Because such diversified manifestations progressively appear with aging, frequent and meticulous follow-ups are required throughout the patient’s lifetime. Consequently, diagnostic confirmation and the precise identification of complementation groups are crucial in predicting clinical symptoms and providing early medical intervention [5,6,7,8].

Throughout time, various diagnostic tools have been developed for XP. Originally, specialized laboratory tests examined the cellular and chromosomal instabilities of XP cells after their exposure to DNA-damaging agents such as ultraviolet (UV) irradiation, chemical carcinogens, or cytotoxic drugs [9,10,11]. In the cellular hypersensitivity test, the colony-forming ability of XP patients’ fibroblasts was measured after UV radiation [10,11]. Similarly, the extent of chromosomal breakage and chromatid aberration was observed in XP cells in response to chemical mutagens. More advanced methods have directly evaluated the repair ability of XP cells [11]. The unscheduled DNA synthesis (UDS) test quantifies newly incorporated nucleotides in XP fibroblasts after UV-induced DNA damage [9]. XP patients only achieve 10–20% of UDS compared to normal subjects. The fusion of identical XP subtype cells leads to UDS recovery, while different ones show no improvement. Other complementation analysis laboratory tests include Western blot and Northern blot, which measure the decrease or absence of specific XP proteins or their mRNA levels [11]. Nevertheless, advancements in molecular science now enable the direct identification of pathogenic mutations in XP genes.

DNA sequencing tools such as single gene testing, targeted gene sequencing (TGS), whole exome sequencing (WES), and whole genome sequencing (WGS) enable rapid and precise XP diagnosis and complementation group analysis [10]. Currently, each method has been recommended for different conditions. When clinical characteristics and regional XP prevalence strongly indicate a certain complementation group, single-gene testing is performed on the associated pathogenic *XP* gene. Alternatively, patients not biased toward a subtype most often perform TGS, which simultaneously sequences multiple pathogenic XP genes. Lastly, when TGS fails to detect mutations in XP-suspected patients, large-scale genome sequencing (WGS or WES) is recommended to discover the genotypes resembling XP clinical features and exclude XP diagnosis.

A 17-year-old female Korean patient (XP115KO) was previously diagnosed with XPC [12]. This patient received regular follow-ups at our hospital for 15 years, from the age of 2 to 17 years old. Unscheduled DNA synthesis and Direct Sanger sequencing for 16 exons of XPC were performed at Kobe University in Japan when she was 6 years old. Direct Sanger sequencing revealed a homozygous nonsense mutation in the *XPC* gene (rs121965088: c.1735C > T, p.Arg579Ter). According to ClinVar [13], a public database of human genome mutations and their relationship to clinical phenotypes, including 10 genetic variants of *XPC*, are currently acknowledged to have pathogenic significance. Among these, rs121965088 is associated with poor prognosis, including multiple skin cancers, internal organ malignancy, and early death [14]. Interestingly, our patient presented with a milder phenotype. Her height and weight were within the normal range, and there were no neurological complications. On metabolic analysis, baseline plasma cortisol and adrenocorticotropic hormone (ACTH) were normal, and no abnormalities were found in the serum glucose, α2-globulins, aspartate transaminase (AST), and alanine transaminase (ALT). First, skin cancer was found at the age of 11 years; however, there have been no additional skin cancers to date (current age, 17 years) and no internal organ malignancies.

The prognosis of XP depends on early diagnosis and strict sun avoidance [15]. Our patient was diagnosed at 6 years of age and practiced diligent sun protection; therefore, her mild phenotype was expected. Additionally, undiscovered genetic variants may have alleviated this phenotype. Essential genes, such as DNA repair genes, are five times more likely to be involved in genetic interactions—with the synergy between two or more genetic mutations creating an unexpected phenotype—than are non-essential genes [16,17,18,19]. Moreover, a “positive” genetic interaction, which reduces the disadvantageous phenotype of a certain mutation, generally involves proteostasis [16], which refers to the intracellular system that maintains protein homeostasis. Since patients with XPC possess mutated endonuclease proteins that fail to recognize and incise damaged DNA, positive genetic interactions may partially restore the DNA repair pathway.

We hypothesized that the genetic interaction—the synergy between two or more mutations creating an unexpected phenotype—may have resulted in a milder phenotype of rs121965088. Hence, we conducted whole-exome sequencing in the patient and her family members to (1) identify pathogenic *XP* mutations affecting the protein-coding regions, (2) detect coexisting mutations in non-pathogenic *XP* genes associated with DNA repair or other genetic disorders, and (3) examine the Mendelian inheritance pattern in the identified biallelic mutations.

## 2. Methods

### 2.1. Subjects and DNA Extraction

We performed the whole-exome sequencing analysis of the samples obtained from the patient and her family members (father, mother, and brother). This family was not related by consanguinity. The study was approved by the institutional review board on 7 July 2017 (No. 2017-05-042-003).

For whole exome sequencing, genomic DNA from the blood was extracted using a Roche DNA Extraction kit (Roche, Indianapolis, IN, USA). Absorbance in 260, 280, and 230 nm for 2 μL of each DNA sample was measured in duplicate using Nanodrop for DNA quality check. The 260/280 ratio was used as an indicator of purity for the DNA samples. For whole exome sequencing, the optimal value of the 260/280 ratio for pure DNA was set to 1.8 or higher, and the experiment was conducted.

### 2.2. Exome Sequencing and Bioinformatics Pipeline

To identify the underlying genetic cause of XPC, the extracted DNA was analyzed using Agilent’s SureSelect XT Human All Exon v5 (Agilent Technologies, Santa Clara, CA, USA).

After sequencing, reads were aligned to an indexed human reference genome (GRCh37/hg19) with the Burrows–Wheeler Aligner (BWA). After alignment, further analysis steps, including (1) data preprocessing and (2) somatic short variant discovery, were used by the Genome Analysis Toolkit (GATK). Additionally, the preparation of the variant calling part was performed on the recalibrated files with detailed steps following GATK best practices. Firstly, Haplotype Caller was used with individual samples in the GVCF mode. Then, all samples were combined in a datastore using the Genomics DB Import function. Next, a joint genotyping of all samples with Genotype GVCFs was performed. To filter out possible false positives, a Variant Quality Score Recalibration was applied to the data. Finally, identified variants were saved in a variant call format. Single nucleotide polymorphisms (SNPs) and indels were annotated using snpEff, which classified the variants.

The functional effects of the resultant variants were predicted using the SNPinfo web server (https://snpinfo.niehs.nih.gov, accessed on 1 January 2022). Structural changes in the XPC protein were visualized using the 3D protein modeling program SWISS-MODEL.

## 3. Results

The nucleotide sequences of XPC patients, parents, and siblings obtained through whole exome sequencing were compared and analyzed. Since XPC disease is an autosomal recessive disease, we focused on finding a homozygous mutation that were not seen in the parents and siblings. As a result of the analysis, the parts showing homozygous mutation in the XPC patients, unlike their families, were found in the olfactory receptor family 2 subfamily T member 35 (*OR2T35*), ALF transcription elongation factor 3 (*AFF3*), TCR gamma alternate reading frame protein (*TARP*), and annexin A7 (*ANXA7*) genes.

Eight biallelic variants, including homozygous in the patient and heterozygous in her parents, were detected (Table 1). Among these, four were found in the *XPC* gene: one nonsense variant (rs121965088: c.1735C > T, p.Arg579Ter) and three silent variants [rs2227998: c.2061G > A, p. Arg687Arg; rs2279017: c.2251-6A > C, intron; rs2607775: c.-27G > C, 5′ untranslated region (UTR)]. When simulated by SWISS-MODEL, rs121965088 caused premature termination during Rad4 protein unit translation, creating a defective and misfolded XPC protein (Figure 1). The remaining four variants were found in non-XP genes: one frameshift variant (rs72452004 of *OR2T35*) and three missense variants (rs202089462 of *AFF3*, rs138027161 of *TARP*, and rs3750575 of *ANXA7*).

## 4. Discussion

In this report, we describe the 15-year follow-up of a Korean girl presenting with a good prognosis despite having the homozygous nonsense mutation of rs12195088 in the XPC gene. The whole-exome sequencing analysis of our patient with XP and her family members revealed four biallelic variants in *XPC* (rs121965088, rs2227998, rs2279017, and rs2607775) and four in non-XP genes (rs72452004 of *OR2T35*, rs202089462 of *AFF3*, rs138027161 of *TARP*, and rs3750575 of *ANXA7*).

No novel ‘pathogenic’ biallelic variants were detected in pathogenic XP genes. The additionally identified three silent mutations of *XPC* (rs2227998, rs2279017, and rs2607775) have already been classified as ‘benign’ variants by the American College of Medical Genetics and Genomics (ACMG) guideline. Nevertheless, since rs2279017 and rs2607775 are located in the intron and 5′UTR region of *XPC*, they may affect RNA splicing and alter the messenger RNA sequence. For instance, the rs2607775 is located in the transcription factor-binding site of the 5′UTR, and previous studies have reported its association with increased cancer risks, including hepatocellular cancer risk, gastric cancer, and colorectal cancer [36,37,38]. The *XPC* gene repairs the DNA damage caused by internal and external factors, and it is known that many cancers occur due to the abnormality of XPC [39]. Therefore, their influence on XP phenotypes, when combined with underlying pathogenic XP variants, needs further study.

Similarly, four biallelic variants located on non-pathogenic XP genes (rs72452004 of *OR2T35*, rs202089462 of *AFF3*, rs138027161 of *TARP*, and rs3750575 of *ANXA7*) had no acknowledged pathogenicity relating to XP or other genetic disorders; they have no reports in ClinVar or meaningful reports in the literature. While rs3750575 was once published to have a higher frequency in Japanese patients with dissecting aortic aneurysms in an exome-wide association study [35], no additional significance has been reported since. Nonetheless, since all four mutations are either frameshift or missense mutations, protein translation—and potentially protein function—are inevitably disturbed. Additionally, proteins encoded from *OR2T35*, *AFF3*, *TARP*, and *ANXA7*, respectively, involve the G-protein-coupled receptor, nuclear transcriptional activator, T cell receptor gamma protein, and calcium-dependent phospholipid binding proteins according to the National Center for Biotechnology Information. Since such proteins are components of major cellular pathways, their concurrence with pathogenic XP variants may influence DNA repair pathways.

Along with pathogenic genotype identification, DNA sequence analysis provides insights into the “genotype-to-phenotype relationship” in XP [40,41,42,43,44]. DNA repair disorders do not possess a one-to-one correlation between molecular genotypes and clinical phenotypes [2,40,45,46]. Mutations in different genes may present with identical clinical manifestations, while genetic variants within a gene may exhibit contrasting phenotypes. Accordingly, XP has shown great diversity in clinical phenotypes within the same complementation groups yet relative uniformity within identical genetic variants in the ClinVar database. In addition, genetic interactions may once again diverge phenotypes within a genetic variant [1,47]. Thus, large-scale genome studies such as WGS and WES enable a comprehensive understanding of genotype–phenotype relations and underlying etiologies of XP.

## 5. Conclusions

In conclusion, we performed a long-term follow-up of an XPC patient with a milder phenotype despite having a mutation (rs12195088) with a poor prognosis. WES was performed on her and her entire family in order to identify possible genetic features. We identified the eight variants related to XPC. Among the eight variants, four variants were located on the *XPC* gene, and three variants (rs2279017, rs2227998, and rs2607775) might be indirectly related to the protein function of XPC. One variant, rs121965088, had been reported to affect the protein of XPC in a previous paper [11]. The remaining four variants, rs72452004, rs202089462, rs138027161, and rs3750575 were located in *OR2T35*, *AFF3*, *TARP*, and *ANXA7* genes, respectively. As the variant of each gene is located in the axon region, it might affect the protein by substituting the amino acid constituting the protein.

The limitation of this study is that biological experiments were not performed. Additional biological studies were needed for each variant, for example, direct changes in the RNA expression or protein structure.

## Figures and Tables

**Figure 1 medicina-59-00699-f001:**
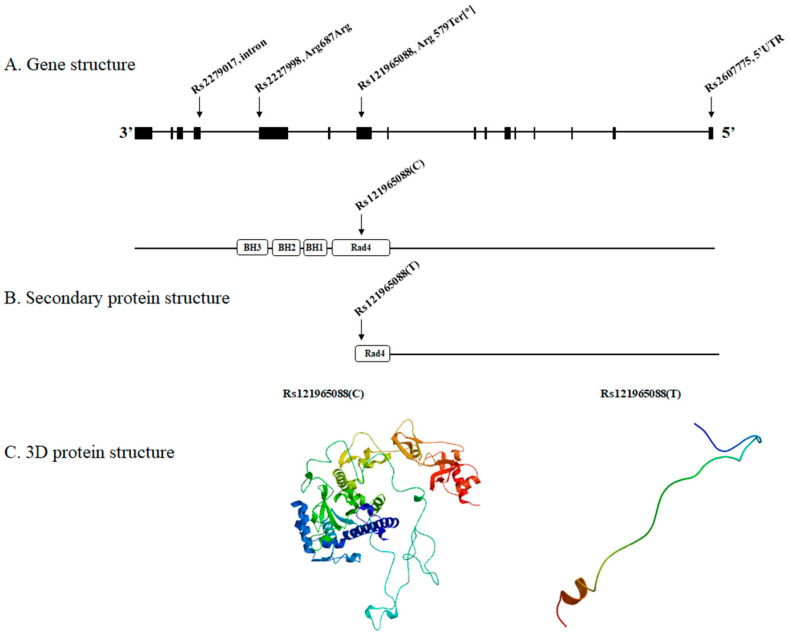
(**A**) The four XPC complex subunit, DNA damage recognition and repair factor (*XPC)* variants (rs121965088, rs2227998, rs2279017 and rs2607775) of our patient located on the *XPC* gene. (**B**) The normal XPC protein is constructed by the consecutive translation of Rad4, BH1, BH2, and BH3 protein units. However, rs121965088 (*) generates a stop codon and results in premature termination during the Rad4 translation. (**C**) Consequently, a defective and misfolded XPC protein is produced. The resultant protein architecture is displayed by the protein 3D modeling program SWISS-MODEL.

**Table 1 medicina-59-00699-t001:** Eight biallelic variants identified in the patient and her family members.

Gene	rs ID	Chromosome Number	Patient	Brother	Father	Mother	Protein Residue	Effect on Amino Acid	Previous Study
** *XPC* **	rs2279017	Chr 3	1/1	0/1	0/1	0/1	Intron	No effect	[20,21,22,23,24,25]
** *XPC* **	rs2227998	Chr 3	1/1	0/1	0/1	0/1	Arg687Arg	Synonymous	[21,26,27,28]
** *XPC* **	rs121965088	Chr 3	1/1	0/1	0/1	0/1	Arg579Ter	Non-synonymous	[14,29]
** *XPC* **	rs2607775	Chr 3	1/1	0/1	0/1	0/1	5′UTR	No effect	[21,30,31,32,33,34]
** *OR2T35* **	rs72452004	Chr 1	1/1	0/1	0/1	0/1	Cys203Ter	Non-synonymous	
** *AFF3* **	rs202089462	Chr 2	1/1	0/1	0/1	0/1	Thr619Ser	Non-synonymous	
** *TARP* **	rs138027161	Chr 7	1/1	0/1	0/1	0/1	Arg100Gly	Non-synonymous	
** *ANXA7* **	rs3750575	Chr 10	1/1	0/1	0/1	0/1	Arg419Gln	Non-synonymous	[35]

XPC complex subunit, DNA damage recognition and repair factor (*XPC*); olfactory receptor family 2 subfamily T member 35 (*OR2T35*); ALF transcription elongation factor 3 (*AFF3*); TCR gamma alternate reading frame protein (*TARP*); annexin A7 (*ANXA7*); untranslated region (UTR); 1/1, homozygous; 0/1, heterozygous.

## Data Availability

Not applicable.

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
