# Peer review of "Whole Exome Sequencing of a Patient with a Milder Phenotype of Xeroderma Pigmentosum Group C"

_medicina, 2023, doi:10.3390/medicina59040699_

Round 1

Reviewer 1 Report (Previous Reviewer 1)

Thank you for your answers to my comments.

The manuscript is now ok for me to be published

Author Response

We appreciate the time and effort you’ve spent on reviewing our manuscript.

Reviewer 2 Report (Previous Reviewer 2)

Ji-In Seo et al., performed whole-exome sequencing in a 17-year-old female Korean patient (XP115KO) previously diagnosed with Xeroderma pigmentosum group C, and discovered potential candidates for genetic interaction with rs121965088. The rs2279017 and rs2607775 of XPC were found to involve mutations in the intron region, which may affect RNA splicing and protein translation. However, the genetic variants of AFF3, TARP, and ANXA7 were all frameshift or missense mutations, affecting the translation and function of the corresponding proteins.

Even though, as the authors stated further research is needed to clarify and understand their function with regard to DNA repair mechanisms, the findings presented in the present manuscript are interesting and relevant for the medical field.

Minor comments:

- Keep consistency in using abbreviation throughout the manuscript.

- Table 1 is introduced at pag.3, line 150, so it should be presented before Figure 1.

- Figure 1C: It would be great if the authors could increase the 3D protein structure.

- Pag. 2, lines 89-90: The authors reported that “First skin cancer was found at the age of 11 years, however there were no additional skin cancer to date (current age, 17 years), and no internal organ malignancies”. However, at pag. 5 lines 172-173, the authors stated that “In this report, we describe 15-year follow-up of a Korean girl presenting with lentiginous pigmentation on sun-exposed areas and basal cell carcinoma (BCC) on the nose”. I would suggest to modify it because it is a little bit confusing.

- Pag.5, line 185: Please, add relative references.

Author Response

Ji-In Seo et al., performed whole-exome sequencing in a 17-year-old female Korean patient (XP115KO) previously diagnosed with Xeroderma pigmentosum group C, and discovered potential candidates for genetic interaction with rs121965088. The rs2279017 and rs2607775 of XPC were found to involve mutations in the intron region, which may affect RNA splicing and protein translation. However, the genetic variants of AFF3, TARP, and ANXA7 were all frameshift or missense mutations, affecting the translation and function of the corresponding proteins.

Even though, as the authors stated further research is needed to clarify and understand their function with regard to DNA repair mechanisms, the findings presented in the present manuscript are interesting and relevant for the medical field.

Minor comments:

- Keep consistency in using abbreviation throughout the manuscript.

Our response: All abbreviations of Xeroderma pigmentosum group C have been changed to XPC. (not using "XP-C")

- Table 1 is introduced at pag.3, line 150, so it should be presented before Figure 1.

Our response: As your suggestion, we reversed the order of Table 1 and Figure 1.

- Figure 1C: It would be great if the authors could increase the 3D protein structure.

Our response: We replaced Figure 1 C with increased 3D protein structure per your suggestion.

- Pag. 2, lines 89-90: The authors reported that “First skin cancer was found at the age of 11 years, however there were no additional skin cancer to date (current age, 17 years), and no internal organ malignancies”. However, at pag. 5 lines 172-173, the authors stated that “In this report, we describe 15-year follow-up of a Korean girl presenting with lentiginous pigmentation on sun-exposed areas and basal cell carcinoma (BCC) on the nose”. I would suggest to modify it because it is a little bit confusing.

Our response: We sincerely apologize for confusing you. We did follow up for 15 years with this patient from her age of 2 to 17 years old. She had BCC in her nose at the age of 11 and had surgery and no other cancer since. We edited ambiguous sentences to the following:

Old manuscript: Introduction

A 17-year-old female Korean patient (XP115KO) was previously diagnosed with XPC [12].

New manuscript: Introduction 

A 17-year-old female Korean patient (XP115KO) was previously diagnosed with XPC [12]. This patient received regular follow-up at our hospital for 15 years from the age of 2 to 17 years old. Unscheduled DNA synthesis and Direct Sanger sequencing for 16 exons of XPC were performed at Kobe University in Japan when she was 6 years old.

Old manuscript: Discussion

In this report, we describe 15-year follow-up of a Korean girl presenting with lentiginous pigmentation on sun-exposed areas and basal cell carcinoma (BCC) on the nose.

New manuscript: Discussion
In this report, we describe 15-year follow-up of a Korean girl presenting with good prognosis despite having the homozygous nonsense mutation of rs12195088 in the XPC gene.

- Pag.5, line 185: Please, add relative references.

Our response: We added the relative reference in the manuscript.

Reviewer 3 Report (Previous Reviewer 3)

The authors considered my concerns but they don’t address it. No experimental evidence has been provided supporting their conclusions. I still consider this manuscript not adequate for publication in the present form.

Main criticisms:

1.      In the abstract (lines 21-23), WES was not carried out to identify the cause of XPC, already identified and published previously. Please remove or rephrase this sentence. Similarly, the Conclusions in the abstract (lines 31-36) don’t have any supporting evidence.

2.      The three additional homozygous variants in XPC are clearly polymorphisms, and any assertion on their effect on the splicing is speculative without any experimental evidence (e.g., RNA analysis).

3.      In my opinion, the proposed 3D model of mutant protein remains speculative and no evidence was provided that this defective and misfolded protein is produced. RNA analysis should be carried out to exclude the role of nonsense mediated RNA decay. In addition, XPC rabbit polyclonal Ab they previously used (PMID 29330851) should be able to detect a truncated protein, as it is developed against the first 16 N-terminus amino acids of XPC protein. This supporting experiment should be carried out.

4.      For the above reasons, the sentences in the “Discussion” section (lines 181-183, lines 196-197, and lines 215-218) are not adequately supported by data.

Author Response

The authors considered my concerns but they don’t address it. No experimental evidence has been provided supporting their conclusions. I still consider this manuscript not adequate for publication in the present form.

Main criticisms:

  1. In the abstract (lines 21-23), WES was not carried out to identify the cause of XPC, already identified and published previously. Please remove or rephrase this sentence. Similarly, the Conclusions in the abstract (lines 31-36) don’t have any supporting evidence.

Our response:

We sincerely apologize for confusing you. We did follow up for 15 years with this patient from her age of 2 to 17 years old. When she was young (2-5 years old), XP was suspected only clinically. At the age of 6, she visited Kobe University in Japan, where she underwent unscheduled DNA synthesis and direct Sanger sequencing of 16 exons of XPC. At the time, she had not undergone whole exome sequencing.

Despite having the homozygous nonsense mutation of rs12195088, which has a poor prognosis among XPCs, she had a good prognosis. That's why we performed whole exome sequencing on her and her entire family this time.

The reason why we conducted WES this time is not to identify the cause of XPC, but to find out why the prognosis is good despite having a mutation (rs12195088) with a poor prognosis. We edited ambiguous sentences to the following:

Old manuscript: Abstract

To identify the underlying genetic cause of XPC, the extracted DNA was analyzed

New manuscript: Abstract

To identify the underlying genetic background of XPC, the extracted DNA was analyzed

Old manuscript: Introduction

A 17-year-old female Korean patient (XP115KO) was previously diagnosed with XPC [12].

New manuscript: Introduction 

A 17-year-old female Korean patient (XP115KO) was previously diagnosed with XPC [12]. This patient received regular follow-up at our hospital for 15 years from the age of 2 to 17 years old. Unscheduled DNA synthesis and Direct Sanger sequencing for 16 exons of XPC were performed at Kobe University in Japan when she was 6 years old.

Old manuscript: Discussion

In this report, we describe 15-year follow-up of a Korean girl presenting with lentiginous pigmentation on sun-exposed areas and basal cell carcinoma (BCC) on the nose.

New manuscript: Discussion
In this report, we describe 15-year follow-up of a Korean girl presenting with good prognosis despite having the homozygous nonsense mutation of rs12195088 in the XPC gene.

  1. The three additional homozygous variants in XPC are clearly polymorphisms, and any assertion on their effect on the splicing is speculative without any experimental evidence (e.g., RNA analysis).

Our response:

I honestly agree with your opinion. biological experiment, for example RNA analysis, would strengthen our results. And It cannot be concluded the variants we have identified have an effect without biological testing. However, additional experiments are not possible, considering the current issues regarding IRB consent, collecting new blood or skin samples, and time limitation for revision. And regarding the topic of "rare skin disease", the issue of this journal, we believe that our study can be used as basic data on the prognosis of XPC patients. Therefore, we addressed this point as limitation of this study in the discussion. 

Since XPC is a very rare skin disease, it will take a long time before more XPC patients' data are accumulated to determine whether the variant shown in our results is really statistically significant. If results similar to ours are reported in other XPC patients in the future, it will be helpful to identify the mechanism of the variant.

New manuscript: Discussion
The limitation of this study is that biological experiments were not performed. Additional biological studies are needed for each variant, for example, direct changes in RNA expression or protein structure.

  1. In my opinion, the proposed 3D model of mutant protein remains speculative and no evidence was provided that this defective and misfolded protein is produced. RNA analysis should be carried out to exclude the role of nonsense mediated RNA decay. In addition, XPC rabbit polyclonal Ab they previously used (PMID 29330851) should be able to detect a truncated protein, as it is developed against the first 16 N-terminus amino acids of XPC protein. This supporting experiment should be carried out.

Our response:

As you comment, it would be great if we could conduct biological experiments. Unfortunately, as of now, we are sorry that additional experiments are not possible due to the IRB consent of our study, collecting new blood or skin samples, and time limitation for revision. However, we think it is meaningful to present some genetic characteristics of the "rare disease" XPC, which is the topic of the special issue. Once again, we are sorry for not presenting the results of biological experiments according to your opinion.

  1. For the above reasons, the sentences in the “Discussion” section (lines 181-183, lines 196-197, and lines 215-218) are not adequately supported by data.

Our response:

I honestly agree with your opinion. It will be able to conclude with certainty when biological experiment results are consistent with our proposed variant. The inappropriately expressed part of the discussion has been corrected and limitations have also been added.

Old manuscript: Discussion

In conclusion, potential candidates for genetic interaction with rs121965088 were found. The rs2279017 and rs2607775 of XPC involve mutations in the intron region, which affect RNA splicing and protein translation. The genetic variants of AFF3, TARP, and ANXA7 are all frameshift or missense mutations, inevitably disturbing the translation and function of the resultant proteins. Further research on their functions in DNA repair pathways may reveal undiscovered cellular relationships within XP.

New manuscript: Discussion
In conclusion, we performed a long term follow-up of an XPC patient with a milder phenotype despite having a mutation (rs12195088) with a poor prognosis. WES was performed on her and her entire family in order to identify possible genetic features. We identified the eight variants related to XPC. Among eight variants, 4 variants were located on the XPC gene, and 3 variants (rs2279017, rs2227998, rs2607775) might be indirectly related to the protein function of XPC. One variant, rs121965088, had been reported to affect the protein of XPC in a previous paper. [11] The remaining four variants, rs72452004, rs202089462, rs138027161, and rs3750575 were located in OR2T35, AFF3, TARP, and ANXA7, respectively. As the variant of each gene is located in the axon region, it might affect the protein by substituting the amino acid constituting the protein.

The limitation of this study is that biological experiments were not performed. Additional biological studies are needed for each variant, for example, direct changes in RNA expression or protein structure.

Round 2

Reviewer 3 Report (Previous Reviewer 3)

I thank authors very much for considering my concerns. However, they were unable to provide the requested experimental evidence supporting their conclusions. For this reason, I still consider this manuscript not adequate for publication in the present form.

I'd like to encourage authors to do any effort to obtaining informed consent and necessary biological samples to carry out the suggested experiments that themselves agree to be necessary.

This manuscript is a resubmission of an earlier submission. The following is a list of the peer review reports and author responses from that submission.

Round 1

Reviewer 1 Report

Manuscript ID: medicina-2160732

Type of manuscript: Article,  Submitted to : Medicina

Title: Whole exome sequencing of a patient with a milder phenotype xeroderma

pigmentosum group C

Authors: Ji-In Seo, Su Kang Kim, Chikako Nishigori, Jung Jin Ahn, Jae Young

Ryu, Junglok Lee, Mu-Hyoung Lee, Ki-Heon Jeong *

Comments to the authors :

-        This is a well written manuscript relating a patient with a milder phenotype xeroderma pigmentosum group C, identified by whome exome sequencing. This manuscript reported an interesting finding about many variants involved in attanuation of the phenotype of Xeroderma Pigmentosum.

-        There are some modifications that will make the manuscript more relevant and complete:

·       The authors identified four biallelic variants in XPC (rs121965088, rs2227998, rs2279017, and rs2607775). These variants should be more discussed.

·       Please add in the table 1, clinical and other molecular findings in patients harboring the four biallelic variants in XPC (rs121965088, rs2227998, rs2279017, and rs2607775).

·       Did these patients had similar phenotype of the reported patient in this manuscript ? Did they have the same mutation in XPC gene? These results should be more discussed in the discussion section.

Reviewer 2 Report

Seo et al, found that a 17-year-old female Korean patient diagnosed with Xeroderma pigmentosum (XP) group C revealed a homozygous nonsense mutation in the XPC gene (rs121965088: c.1735C>T, p.Arg579Ter) that generally is associated with a poor prognosis. However, the patient had only a mild phenotype. The authors performed whole-exome sequencing to detect coexisting mutations in the patient and her family members and discovered eight potential candidates for genetic interaction.

The manuscript is well written and organized, it is interesting and has a high clinical significance.

Even though, further research is required to characterized the functions of these genetic variants in the context of DNA repair and XP, I would encourage the authors to provide with any information related to metabolic analysis/changes if available.

Reviewer 3 Report

By WES, authors investigate a Korean patient with xeroderma pigmentosum group C and milder phenotype, in which they had already identified the causative variant in XPC. The aim is to identify additional genetic variants that could justify the observed phenotype variability. In my opinion, this manuscript present serious limitations and authors’ conclusions are not adequately supported by data.

 Main criticisms:

11. By WES, authors identify three additional homozygous variants in XPC that are clearly polymorphisms, on which they speculate on a possible effect on splicing. No evidence is provided. In addition, they also report homozygous polymorphic variants in four genes not related to xeroderma pigmentosum or XPC. Similarly, no evidence is provided on their contribute to phenotype variability. Reported variants are heterozygous in unaffected parents and brother. The homozygosity in affected patient should suggest consanguinity in this family, but no information were provided about this. As this case has been previously investigated (PMID 29330851) by Sanger sequencing of all XPC gene, I don’t understand why additional XPC variants were not previously reported.

22.  Authors develop a 3D model of mutant protein (figure 1), speculating on the translation of a defective and misfolded protein. Their observations are clearly in contrast with previously reported data (PMID 29330851), in which WB analysis on patient’s fibroblast clearly demonstrated the absence of XPC protein, possibly due to nonsense mediated decay.

33. Concerning rs2607775 polymorphism in 5’UTR of XPC, authors cite previous studies that reported its association with an increased cancer risk. They speculate on its possible contribute to XP phenotype, forgetting the presence of a nonsense variant in the same gene.

Minor criticisms:

In the text “homogeneous” should be “homozygous” and “heterogeneous” should be “heterozygous”.